# Advanced Glycation End Products and Nitrosamines in Sausages Influenced by Processing Parameters, Food Additives and Fat during Thermal Processing

**DOI:** 10.3390/foods12020394

**Published:** 2023-01-13

**Authors:** Jingnan Lu, Mingyu Li, Mingyue Shen, Jianhua Xie, Mingyong Xie

**Affiliations:** State Key Laboratory of Food Science and Technology, Nanchang University, Nanchang 330047, China

**Keywords:** advanced glycation end products, nitrosamines, α-dicarbonyl compounds, sausages

## Abstract

Advanced glycation end products (AGEs) and nitrosamines (NAs) in sausage are associated with pathogenic and carcinogenic risks. However, the multiple reaction parameters affecting the production of AGEs and NAs during sausage processing remain unclear. This experiment evaluated the effects of processing parameters, food additives and fat ratios on the formation of AGEs and NAs in sausages. The results showed a 2–3-fold increase in N^ε^-(carboxymethyl)lysine (CML) and N^ε^-(carboxyethyl)lysine (CEL) when the sausage processing temperature was increased from 90 °C to 130 °C, and N-nitrosodimethylamine (NDEA) increased from 3.68 ng/g to 6.41 ng/g. The addition of salt inhibited the formation of AGEs and NAs, and the inhibitory ability of 2 g/100 g of salt was 63.6% for CML and 36.5% for N-nitrosodimethylamine (NDMA). The addition of 10 mg/kg nitrite to sausages reduced CML formation by 43.9%, however, nitrite had a significant contribution to the formation of NAs. The addition of fat only slightly contributed to the production of CML. In addition, the relationship between α-dicarbonyl compounds and the formation of AGEs was investigated by measuring the changes in α-dicarbonyl compounds in sausages. The results showed two trends of AGEs and α-dicarbonyl compounds: AGEs increased with the increase in α-dicarbonyl compounds and AGE level increased but α-dicarbonyl compound level decreased.

## 1. Introduction

Meat and meat products are an important component of diets around the world, and sausages make up a high proportion of these meat products. Sausages are rich in protein and fat and are generally subjected to a more rigorous heat treatment than foods of plant origin. Heat treatment results in the production of some harmful products, such as advanced glycation end products (AGEs) [1]. AGEs are formed through non-enzymatic reactions between free amino groups from amino acids, peptides or proteins and active carbonyl groups from reducing sugars and are products of the advanced stage of the Maillard reaction [2,3]. Some epidemiological studies have shown that consumption of AGEs may contribute to metabolic chronic diseases including diabetes, renal disorders and Alzheimer’s disease [4]. N^ɛ^-carboxymethyl lysine (CML) and N^ɛ^-carboxyethyllysine (CEL) are widely used as markers for indicating AGEs in processed foods [5]. The α-dicarbonyl compounds including methylglyoxal (MGO), glyoxal (GO) and 3-deoxyglucuronide (3-DG) have been identified as intermediate products of the Maillard reaction [6,7], and they are important precursors for the formation of AGEs. α-Dicarbonyl compounds are produced in various ways: α-dicarbonyl compounds are formed by oxidative cleavage of reducing sugars; Amadori compounds form α-dicarbonyl compounds through a series of dehydration, oxidation and rearrangement reactions; and fat peroxidation directly produces α-dicarbonyl compounds [8].

The reaction of nitrite with secondary amines during sausage processing produces N-nitrosamines (NAs), which are aliphatic or aromatic derivatives of secondary amines. NAs contain both volatile and non-volatile forms, and volatile NAs (VNAs) are considered to be more carcinogenic. N-nitrosodimethylamine (NDMA) and N-nitrosodiethylamine (NDEA) are the representative VNAs. Epidemiological study suggests a potential relationship between NAs and cancer risk, and a survey of the total intake of VNAs in Swedish food found that about 80% was caused by meat and meat products [9]. Due to the hazardous nature of NAs, many countries set limits on the amount of NAs in meat products. Canada specifies the level of NDMA at 10 μg/kg bw/day [10]. China sets a limit of 3 μg/kg for NDMA in meat products [11].

Many factors influence the formation of AGEs and NAs in sausages, such as processing conditions, fat content and the amount of nitrite [12,13]. Meat products that are processed at high temperatures, such as frying and deep-frying, form high levels of AGEs and NAs [14]. Chen et al. [15] reported that deep-fried meat products contain higher levels of AGEs. Food additives such as nitrite and NaCl are widely used in sausages for flavoring and microbial inhibition [16], which have been shown to affect the formation of AGEs and NAs in various meat products. Salt affects protein breakdown and lipid oxidation in sausages, adding a special flavor. Nitrite develops color by forming NO that reacts with myoglobin to form nitrosomyoglobin, providing a stable red or pink color to the sausage. Nitrite also acts as an antioxidant, inhibits the growth of certain microorganisms and provides a pleasant curing flavor [17]. However, nitrite is an important factor in the formation of NAs and can significantly promote the formation of NAs [18].

There are no reports on the simultaneous study of AGE and NA formation in sausages, and the influence of processing conditions on the contents of AGEs and NAs in sausages is still unknown. AGEs and NAs in sausages are influenced by multiple factors, and simulated systems and single-study systems have not fully reflected the formation patterns of AGEs and NAs. The aim of this study was to construct a realistic sausage processing system and investigate the effects of processing conditions, additives and fat on the formation patterns of AGEs and NAs. Meanwhile, the formation of AGEs and α-dicarbonyl compounds was monitored dynamically, and the correlation of AGEs and NAs with physicochemical indicators (fat, protein, water, etc.) was further evaluated.

## 2. Materials and Methods

### 2.1. Reagents

Four AGE standards, CML (>98%), CEL (>98%) and its isotopically labeled compounds d_4_-CML (>98%) and d_4_-CEL (>98%), were purchased from Toronto Research Chemicals Inc. (Toronto, ON, Canada). Glyoxal (GO) solution (approx. 40%), methylglyoxal (MGO) solution (approx. 40%), 3-Deoxyglucosone (3-DG) (95%) and o-phenylenediamine (OPD) (99.5%) were obtained from Sigma-Aldrich (Steinheim, Germany). NDMA and NDEA were obtained from J&K Scientific Ltd. (Beijing, China). Oasis MCX solid-phase extraction cartridges (6 mL, 150 mg) were purchased from Waters Co. (Milford, MA, USA). Sodium borohydride, sodium hydroxide, sodium borate, boric acid, disodium hydrogen phosphate, ascorbic acid and sodium dihydrogen phosphate were purchased from Shanghai Aladdin biochemical technology Co., Ltd. (Shanghai, China). Acetonitrile and methanol were purchased Merck Company (Darmstadt, Germany). Hydrochloric acid (37%), petroleum ether and hexane were purchased Xilong Scientific Co., Ltd. (Guangdong, China). Bond Elut QuEChERS extraction package and QuEChERS matrix dispersion purification package components were obtained from Agilent Technologies Co., Ltd. (Santa Clara, CA, USA). Raw pork was purchased from Rainbow Supermarket (Nanchang, China). Collagen enteric coating was purchased from North Products Food Co (Zhaozhuang, China).

### 2.2. Preparation of Sausage Samples

Fresh pork hind leg meat was purchased from Tianhong Supermarket (Nanchang, China). The pork was stripped of visible connective tissue and fat, and the lean meat was cut into small pieces and ground using an electric meat grinder. Fatty meat was also churned by electric grinding for the experiment. The minced meat was mixed with salt, nitrite and fat according to Table 1. Each group of cured meat was divided into 3 portions of approximately 100 g each. After curing at 4 °C for 6 h, the cured meat was filled into collagen casings to form pork sausages. Sausage weight was 60 ± 10 g, diameter was 1.8 cm and length was 10 cm (due to loss of minced meat due to apparatus during production, one sausage was made per 100 g of sausage). The pork sausages produced were processed at different temperatures and time conditions (Table 1). The experiment was repeated three times for all groups of treated pork sausages

### 2.3. Physicochemical and Color VALUE analysis

Fat, protein, moisture and nitrite residue were determined according to AOAC procedures [19]. The color values were determined using a Sann colorimeter NR100 (Qingdao, China). The samples were spread out evenly and the color values were measured at 6 different positions using a hand-held colorimeter, resulting in an average of L*, a* and b* [20].

### 2.4. AGE and α-Dicarbonyl Compound Analysis

#### 2.4.1. Extraction and Purification of AGEs and α-Dicarbonyl Compounds

The detection method of AGEs was modified according to [21,22]. Referring to the method of [22], an OPD derivatization method was used to determine the content of α-dicarbonyl compounds in sausage. Sausage sample (0.5 ± 0.05 g) was placed in a thick-wall pressure-resistant glass tube and mixed with 5 mL water and vortexed for 10 min. Then, 5 mL of acetonitrile was added, and the sample was vortexed for 10 min, shaken for 20 min and centrifuged through the membrane. Then, 500 μL supernatant was added with 500 μL of 0.2% OPD aqueous solution (ready to prepare), derivatized at 37 °C for 4 h, passed through 0.22 μm organic filter membrane and analyzed by UPLC-MS/MS.

#### 2.4.2. Determination by UPLC-MS/MS

The content of AGEs and α-dicarbonyl compounds in sausages was determined using 1290–6460 UPLC–MS/MS (Agilent Technologies, Inc.) equipped with Synergi Hydro–RP 80 Å UPLC column (2 × 250 mm, 4 μm; American Phenomenex Scientific Instruments Company). Detection was performed using MS operated in electrospray ionization (ESI) positive mode with multiple reaction monitoring (MRM). The mobile phase of AGEs was formic acid water (100%), the injection volume was 1 μL, the mobile phase rate was 0.15 mL/min, the column temperature was 25 °C and the sample analysis time was 9 min. The mobile phase of the α-dicarbonyl compounds was 0.1% formic acid water (B) and acetonitrile (A) (*v*/*v*), the injection volume was 5.0 μL, the mobile phase rate was 0.3 mL/min, the column temperature was 30 °C and the sample analysis time was 15 min.

### 2.5. NDMA and NDEA Analysis

#### 2.5.1. Extraction and Purification of NDMA and NDEA

An amount of 10 g of crushed sample was added to 10 mL of acetonitrile extraction solvent, vortexed and then frozen in a −20 °C refrigerator for 30 min. Then, 2 ceramic homogenizers were added in addition to bond elut QuEChERS extraction salt, and the solution was shake rapidly for 30 s and centrifuged at 8000 r/min at 0 °C for 10 min. Next, 5 mL of supernatant was added to the 15 mL bond elut QuEChERS substrate dispersion purification tube, vortexed for 1 min and centrifuged at low temperature 0 °C for 10 min at 8000 r/min. An amount of 5 mL of supernatant was aspirated for nitrogen blowing, but not completely blown dry [23]. Finally, 1 mL of dichloromethane was added for re-solubilization and filtered through a 0.22 μm filter membrane into a sample vial for analysis.

#### 2.5.2. Determination by GC-MS

The contents of NDMA and NDEA in sausages were determined using an Agilent 7890B gas chromatograph and an Agilent 7000D triple quadrupole mass spectrometer (GC-MS/MS, Agilent Technologies Inc., Santa Clara, CA, USA) equipped with a DB-35 ms column (30 m × 0.25 mm × 0.25 μm; Agilent Technologies Inc., Santa Clara, CA, USA).

### 2.6. Statistical Analysis

Each index measurement was repeated three times. The experimental data were statistically analyzed using SPSS 17.0 software and the data were analyzed for significance by ANOVA monofactor analysis. Duncan’s analysis was used to compare significant differences between groups, with *p* < 0.05 being the test of significance, and pre-experimental data were analyzed using mean ± standard error. Graphical processing was carried out using GraphPad 8.0 software. In addition, correlation analysis was performed between NAs, AGEs and related factors, and the correlation analyses were processed by Origin.

## 3. Results and Discussion

### 3.1. Physicochemical Characteristics and Color Value

As shown in Table 2, the fat content gradually decreased as the processing temperature and time increased. The fat content was 3.07% when baked at 70 °C for 2 h, and decreased to 0.55% at a processing temperature of 130 °C. The fat content of the sausages decreased from 2.77% to 0.70% after 4 h of processing, and this was mainly due to the reduction in fat content as the fat drips out during the cooking process at high temperatures and for long processing times. Due to the high-temperature and low-humidity environment, the moisture content of the sausages also decreased continuously after roasting. The moisture content was 64.2% at 70 °C and reduced to 55.48% at 130 °C. The moisture content of the sausage was 52.72% after 4 h of roasting, which was significantly lower than for 1 h of roasting (66.42%). The addition of salt increased the moisture, and the moisture increased from 52.95% to 59.99% when salt was added at 2 g/100 g (Table 2). In agreement with the present results, Ruedt et al. [24] concluded that salt treatment significantly (*p* < 0.05) increased the water holding capacity of pork. The addition of salt slowed the migration of water within the muscle, which enhanced the water retention capacity of the sausage and reduced the steam loss of the sausage. The moisture content of the sausage decreased from 64.37% to 59.48% after 15 mg/100 g of nitrite was added, which could be due to the nitrite curing effect. Gómez et al. [25] suggested that coagulation of meat fibers occurs when nitrite and sodium ions migrate rapidly and undergo a dehydration reaction. It has also been suggested that osmotic dehydration of raw meat caused by the addition of nitrite leads to a decrease in sausage moisture [26].

Temperature and time significantly increased the protein content in the sausages, which may be due to the concentration effect caused by moisture loss. As can be seen from Table 2, the increase in baking time and temperature resulted in a decrease in nitrite residues (NR). This was mainly due to the fact that roasting accelerated the disproportionation reaction of nitrite and the degradation reaction of nitrite. At the same time, nitrite penetrated into the meat through water, and the reduction in water reduced its migration rate [27].

The color value can reflect the degree of the Maillard reaction. As seen in Table 3, a* increased significantly with the increasing processing temperature and time. From Table 3, it can be seen that the L* and a* values in sausages increased significantly after nitrite addition. The addition of nitrite promoted the oxidation of myoglobin, and this result was in agreement with Deng et al. [28]. Previous studies have concluded that low concentrations of salt can increase the L* of sausages and high concentrations of salt can decrease the L* [24]; our results indicated that L* was 62.23 at 2% salt addition, but began to decrease as salt addition increased to 5%.

### 3.2. Effect of Cooking Temperature and Time

#### 3.2.1. CML and CEL

Temperature and time are important processing parameters in the thermal processing of sausages, so this experiment assessed the effect of processing temperature and time on CML and CEL formation (Figure 1). The levels of CML and CEL were almost unchanged at 70 °C and 90 °C, while they increased rapidly when the temperature was 110 °C. The levels of CML and CEL were 2.88 μg/g and 8.12 μg/g at 110 °C and increased to 6.58 μg/g and 16.32 μg/g at 130 °C, respectively: the levels of CML and CEL increased by about 2-fold (Figure 1A). The results were similar to results presented in a previously published study [29], which also found that the levels of CML and CEL increased slowly at lower temperatures, and increased rapidly when the temperature was higher than 110 °C. The levels of CML and CEL in heat-processed meat were much higher than that of raw meat [15]. CML and CEL levels are more pronounced at higher temperatures, as high-temperature conditions (121 °C) result in peptide bond breakage, protein structure unfolding and exposure to reactive amino acids. These changes in protein structure promote the occurrence of the Maillard reaction, which facilitates the production of CML and CEL [30]. It has also been proposed that when the heating temperature was lower the inner layer of the sausage was less heated and the degree of fat oxidation or the rate of Maillard reaction decreased, so the CML was more concentrated in the outer layer [15].

Figure 1B shows the correlation between the levels of CML and CEL and time. It was found that the CML increased from 1.29 μg/g to 6.22 μg/g and the CEL level increased from 5.24 μg/g to 9.13 μg/g with a 4 h long processing time when compared with the sausages roasted for 1 h. The levels of CML and CEL gradually increased, and the level of CML changed faster than CEL. Sun et al. [29] also investigated the effect of heating times (from 0 to 60 min) on the levels of bound CML and CEL and found that the contents of CML and CEL in beef gradually increased with increasing heating time. The rate of CML formation in ground beef was faster than CEL, and their formation followed the law of zero-level reaction, which was consistent with the results of our study. Liu et al. compared the levels of AGEs in sturgeon fried for 0, 2, 5, 8 and 11 min and found that frying time significantly increased the formation of CML and CEL [31]. Due to the prolonged processing time, the internal moisture migration and volatilization of sausages, the heat inside the sausage increased, the Maillard reaction and lipid oxidation also increased.

#### 3.2.2. α-Dicarbonyl Compounds

The α-dicarbonyl compounds in sausages were affected by temperature: 3-DG increased from 0.61 μg/g to 1.11 μg/g, GO content increased from 0.58 μg/g to 1.61 μg/g and MGO increased from 3.14 μg/g to 7.13 μg/g (Figure 2A). The results show that the levels of α-dicarbonyl compounds were elevated and accelerated under high-temperature conditions, which may be the result of the combined effect of the Maillard reaction and lipid oxidation. The accumulation of α-dicarbonyl compounds accelerated the formation of CML and CEL. Eggen and Glomb [32] studied the changes in GO and MGO levels in grilled pork. The MGO in raw meat was 0.7 mg/kg and increased significantly to 5.6 mg/kg after grilling. Moreover, the production of MGO was significantly higher than that of GO, which was mainly due to the fragmentation of triphosphate during the heat treatment to produce a large amount of MGO [32].

The content of MGO decreased from 7.92 μg/g to 3.55 μg/g after 2 h of heating (Figure 2B). It was previously reported that the production of α-dicarbonyl compounds was promoted by heating time, as the content of MGO increased from 2.03 mg/kg to 2.89 mg/kg when fish oil (tuna, etc.) was heated at 60 °C for 7 days [33]. However, the results of this test showed that the content of α-dicarbonyl compounds decreased under the condition of increasing heating time, but the content of CML and CEL still increased. Lipid oxidation was an important pathway to produce α-dicarbonyl compounds in meat products. Lipid oxidation was less affected by time, but heating time can promote the formation of AGEs. Therefore, the rate of α-dicarbonyl compound formation was lower than the rate of consumption, and most of the α-dicarbonyl compounds were converted to CML and CEL.

#### 3.2.3. NDMA and NDEA

The effects of processing temperature and time of sausage on the formation of NDMA and NDEA are shown in Figure 3. The level of NDMA was 3.68 ng/g at 70 °C and increased to 6.41 ng/g when the temperature increased to 130 °C, which indicated that processing temperature can promote the growth of NDMA. Heat treatment caused an increase in the concentration of NAs, and the high temperature provided the activation energy required for nitrosation [34]. Drabik-Markiewicz et al. [35] found that the NDMA content of cured pork increased continuously with increasing temperature at 85–220 °C. In addition, processing time was found to have an effect on the formation of NDMA in this experiment, but the level of NDEA was not affected by processing time. The sausages were roasted for 4 h, and the NDMA rose significantly from 4.51 ng/g to 5.53 ng/g.

### 3.3. Effect of Additives

#### 3.3.1. CML and CEL

The effects of salt and nitrite concentrations on the formation of AGEs are shown in Figure 4. The CML level was significantly suppressed with increasing salt addition. The levels of CML and CEL in the group without salt addition were 6.52 μg/g and 6.21 μg/g, and the level of CML was 2.37 μg/g when salt was 2 g/100 g. Salt (2 g/100 g) addition also had an inhibitory effect on CEL, which decreased from 6.56 μg/g to 5.54 μg/g. It has been reported that NaCl can accelerate the lipid oxidation of meat [16], and Tan and Shelef [36] found that NaCl accelerated fat oxidation in pork and beef under low-temperature storage conditions. Kanner [37] reported that NaCl can release iron ions and has pro-oxidant activity. Furthermore, a high concentration of salt resulted in a high degree of protein carbonylation, as the lysine residue ε-NH2 on the protein surface was more susceptible to attack, which led to an increase in protein carbonyl content [38]. The addition of NaCl also affected the structure of myofibrillar proteins [39], thus promoting the production of AGEs. However, Figure 4A shows that the addition of salt can inhibit the formation of CML, and this may be attributed to the fact that salt caused the muscle fibers to swell with water, which improved the capacity of the meat to trap water, and the meat was less likely to lose water during heating.

The sausages were spiked with 0, 5, 10 and 15 mg/100 g nitrite and the effect of nitrite on AGEs in the sausages is shown in Figure 4B. CML and CEL were 4.21 μg/g and 5.54 μg/g, respectively, when the sausages were not supplemented with nitrite. When the concentration of nitrite was 10 mg/100g, CML was reduced by 43.9%, however, nitrite had no inhibitory effect on CEL. Berardo et al. [40] reported that nitrite had a strong inhibitory effect on lipid oxidation, and the addition of nitrite may have inhibited the production of AGEs. Nitrite inhibited CML produced by lipid oxidation of meat [38]. Feng et al. [38] found that nitrite could inhibit protein oxidation and lipid oxidation in ham to reduce the production of AGEs. However, in contrast to the results of this experiment, previous studies believed that nitrite was more effective in inhibiting CEL: it was found that 10 mg/100 g nitrite could inhibit CEL by 17–33% [17]. Nitrite in sausages formed large amounts of NO and NO2, which were more involved in oxidation reactions than lipids and proteins [17]. This may be the mechanism of nitrite that inhibited CML and CEL formation. Meanwhile, nitrite can form stable complexes with iron ions, which reduced the release of heme iron and inhibited the oxidative catalysis of free iron, thus reduced the production of AGEs [41].

#### 3.3.2. α-Dicarbonyl Compounds

As can be seen in Figure 5, three α-dicarbonyl compounds also showed different degrees of inhibition by salt. The best inhibition was observed for 3-DG and MGO when salt was 1 g/100 g, as the levels of 3-DG and MGO decreased by 59% and 67.8%, respectively. The salt inhibited the formation of highly reactive carbonyl compounds from fat peroxidation, and the formation of α-dicarbonyl compounds was inhibited. Rhee et al. [42] reported that NaCl activated lipid oxidation at low concentrations, but inhibited lipid oxidation at concentrations above 2% in minced meat. However, it was reported that the addition of NaCl mainly promoted the formation of malondialdehyde (MA) and had no significant effect on the GO content [43]. Nitrite had an inhibitory effect on GO and MGO, and the best inhibition was achieved with the addition of 5 mg/100 g of nitrite. GO decreased from 0.68 μg/g to 0.34 μg/g and MGO level decreased from 5.53 μg/g to 3.86 μg/g. The most significant inhibition effect was observed when 3-DG decreased from 0.91 μg/g to 0.14 μg/g with the nitrite addition of 15 mg/100 g (*p* < 0.05). Bonifacie et al. [44] found that nitrite has a strong antioxidant capacity in dry-cured fermented sausages, as the addition of 80 ppm nitrite to dry-cured fermented sausages reduced lipid peroxidation. The reduction in nitrite salts prevented or slowed down certain intermediate reactions of the Maillard reaction [45]. The results showed that salt and nitrite reduced CML and CEL mainly by inhibiting lipid oxidation and reducing the production of α-dicarbonyl compounds in sausages.

#### 3.3.3. NDMA and NDEA

Salt and nitrite are indispensable additives in the processing of sausages, and the effects of these additives on the production of NAs are shown in Figure 6. Salt had a significant inhibitory effect on NDMA and NDEA, with 5.48 ng/g and 2.01 ng/g of NDMA and NDEA, respectively, in the no salt addition group (Figure 6A). The inhibition ability of NDMA was 36.5% when salt was added at 2 g/100 g, and when 1 g/100 g of salt was added to the sausage the level of NDEA was reduced to 1.56 ng/g. The secondary amines produced by proteolysis were important precursors for the formation of NAs, and the addition of NaCl reduced the decomposition of proteins. At the same time, sodium chloride deprived microbial cells of water and became inactive, which reduced the NAs produced by microbial activity [38]. The levels of NDMA and NDEA increased with the addition of nitrite at 0, 5, 10 and 15 mg/100 g. The levels of NDMA and NDEA did not increase significantly at lower concentrations of nitrite. When 15 mg/100 g nitrite was added, NDMA increased from 2.32 ng/g to 3.48 ng/g and NDEA increased from 2.66 ng/g to 3.57 ng/g. Drabik-Markiewicz et al. [35] added 0–480 mg/kg of nitrite to cured pork, and the levels of NAs increased from ND (not detected) to 0.441 µg/mL. Herrmann et al. [46] added different concentrations of nitrite (0–350 mg/kg) to investigate the effect of nitrite on NA in meat products, and it was found that increasing the level of nitrite had a limited effect on the level of NDMA. NDMA began to decrease at additions above 150 mg/kg, and this was mainly due to the complete consumption of amines produced by protein breakdown.

### 3.4. Effect of Fat

In addition to the traditional Maillard reaction, lipid peroxidation is often used as an important pathway affecting CML and CEL in meat products [47]. To investigate the effect of fat percentage on AGEs and NAs, this experiment explored the levels of AGEs and NAs in sausages with different fat percentages (0, 10%, 20% and 30%) (Figure 7). The levels of CML and CEL were 2.36 μg/g and 5.54 μg/g when no fat was added. When the percentage of fat was 20%, the level of CML increased from 2.36 μg/g to 3.97 μg/g, and with the increase in fat content, a stable trend was observed. The addition of fat promoted the production of CEL, but there was no significant effect on the level of CEL at different additions. This may be related to hydroxyl radicals (·OH) and lipid peroxyl radicals (LOO-) by lipid oxidation, which can promote the generation of CML in the Maillard reaction model system [48,49]. However, Sun et al. [50] found that the addition of different levels of beef fat during thermal processing had no significant effect on either free AGEs or bound AGEs.

Lipid addition promoted the formation of GO and MGO, with little effect on 3-DG, and a 30% fat addition was able to promote an increase in MGO from 3.53 μg/g to 5.97 μg/g (Figure 7B). The accumulation of MGO was caused by lipid degradation during processing and storage [33], and previous studies have suggested that lipid oxidation during high-temperature heating of meat when it contains fat produces more GO and MGO, and that the degree of lipid peroxidation reacts more rapidly under high-temperature conditions (e.g., frying) [51]. Previous studies found that butter yielded about three times more MGO than GO in 200 °C processing [52].

The results of this test showed no effect of different fat contents on the formation of NDMA and NDEA (Figure 7C). However, previous studies have suggested that nitrosating reagents generated by the reaction of fat with nitrite promote the production of NAs [53]. Mottram et al. [54] separated fatty and lean bacon after frying and found that the former contained significantly higher levels of NAs than the latter. Previous studies [54] divided lamb into fat and fat-free groups and showed that NAs were high in the group containing fat. The lower heating temperature of this experiment than previous studies may have explained the difference between the results of this experiment and previous ones, which is one possible explanation for the poor effect of fat on the production of NAs. Meanwhile, NAs can be lost from the meat through steam distillation during heating.

### 3.5. Correlations between AGEs, NAs and α-Dicarbonyl Compounds

This experiment investigated the correlation of AGEs and NAs in sausages with α-dicarbonyl compounds, fat, protein, moisture, nitrite residues, L*, a* and b* (Figure 8). CML was correlated with 3-DG (r = 0.586, *p* < 0.05), with a significant negative correlation with moisture (r = − 0.790, *p* < 0.01), and also showed a negative correlation with nitrite residues (r = − 0.516, *p* < 0.05). CEL showed significant positive correlations with GO (r = 0.740, *p* < 0.01), protein (r = 0.783, *p* < 0.01) and b* (r = 0.639, *p* < 0.01) and significant negative correlations with fat (r = − 0.644, *p* < 0.01). It was previously reported that 3-DG, GO and MGO can be important precursors by reacting with lysine and arginine to produce CML and CEL [32]. Han et al. [49] found that hydroxyl radicals generated by lipid oxidation differentially promoted the production of CML from lysine and GO in glycosylation reactions by building a reduced-fat sugar-lysine model. The α-dicarbonyl compounds produced by lipid peroxidation in sausage may be an important pathway for the generation of AGEs. The α-dicarbonyl compounds were not only Maillard reaction intermediates, but also direct precursor substances for CML and CEL, effectively promoting CML and CEL reaction rates [55,56,57].

NDMA showed positive correlation with 3-DG (r = 0.579, *p* < 0.05), GO (r = 0.581, *p* < 0.05) and protein (r = 0.583, *p* < 0.05), which was mainly due to the fact that proteolysis and fat oxidation are important influencing factors in the formation of NAs [58]. NDMA had a significant negative correlation with moisture (r = −0.707, *p* < 0.01). Microbial activity is weaker at lower sausage moisture levels and less amines are produced by microbial activity [38]. There was no significant correlation between NAs and nitrite residues, and previous studies have also demonstrated that nitrite residue levels do not correlate with the production of NAs [59]. This was due to the fact that nitrite is strongly reduced during processing, so the nitrite residue does not reflect the true amount added.

## 4. Conclusions

Endogenous AGEs and NAs have been recognized as important factors affecting human health, and the formation mechanism of high levels of AGEs and NAs in sausages has not been clearly explored. In this experiment, we investigated the factors influencing the formation of AGEs and NAs in sausage by establishing a real processing system. The results showed that the important factors for the production of AGEs in sausages were processing temperature and time. CML and CEL did not increase significantly at lower temperatures, but increased rapidly above 110 °C. The effect of temperature and nitrite caused significant growth of NAs. Meanwhile, salt was able to inhibit the production of AGEs, NAs and α-dicarbonyl compounds simultaneously. There were two trends in α-dicarbonyl compounds and AGEs: AGEs increased with α-dicarbonyl compounds and AGEs increased significantly but α-dicarbonyl compound levels decreased, which was mainly due to the faster formation of AGEs from α-dicarbonyl compounds. These data provide research ideas for future studies, such as low-temperature slow-roasting heating conditions, to find suitable additive substitutes and additive concentrations that can effectively reduce AGE and NA levels.

## Figures and Tables

**Figure 1 foods-12-00394-f001:**
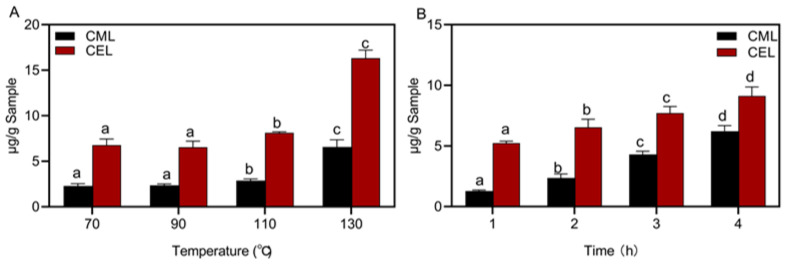
Effect of processing temperature and time on CML and CEL. (**A**) temperature; (**B**) time. Different letters above the bars indicate significant differences (n = 3, *p* < 0.05). N^ɛ^-carboxymethyl lysine (CML) and N^ɛ^-carboxyethyllysine (CEL).

**Figure 2 foods-12-00394-f002:**
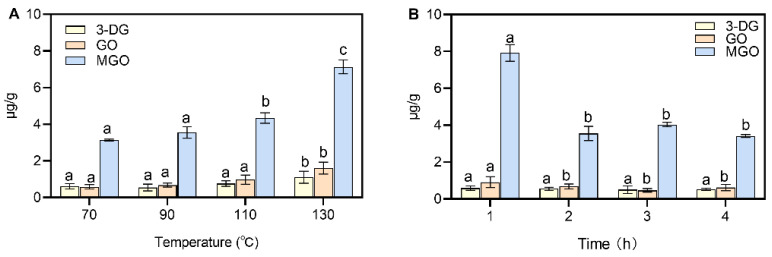
Effect of processing temperature and time on α-dicarbonyl compounds. (**A**) temperature; (**B**) time. Different letters above the bars indicate significant differences (n = 3, *p* < 0.05). Methylglyoxal (MGO), glyoxal (GO) and 3-deoxyglucuronide (3-DG).

**Figure 3 foods-12-00394-f003:**
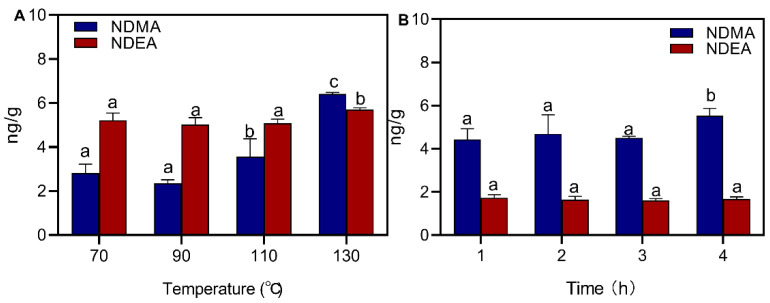
Effect of processing temperature and time on NDMA and NDEA. (**A**) temperature; (**B**) time. Different letters above the bars indicate significant differences (n = 3, *p* < 0.05). N-nitrosodimethylamine (NDMA) and N-nitrosodiethylamine (NDEA).

**Figure 4 foods-12-00394-f004:**
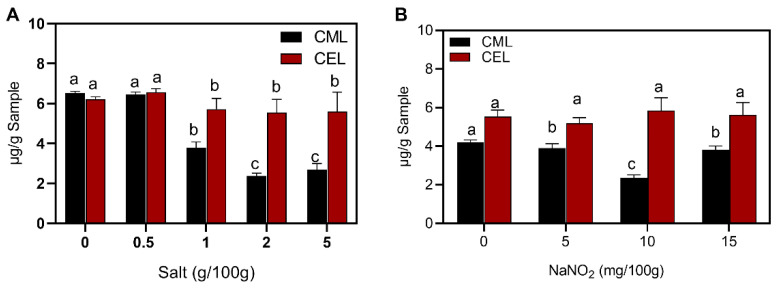
Effect of additive on CML and CEL: (**A**) salt; (**B**) nitrite. Different letters above the bars indicate significant differences (n = 3, *p* < 0.05). N^ɛ^-carboxymethyl lysine (CML) and N^ɛ^-carboxyethyllysine (CEL).

**Figure 5 foods-12-00394-f005:**
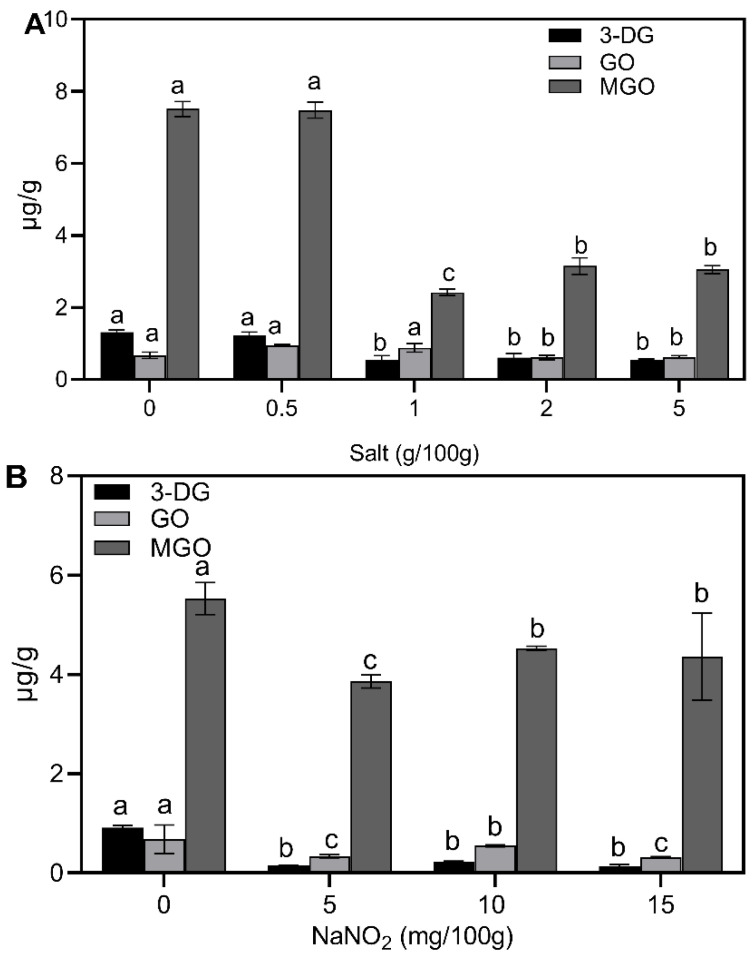
Effect of additive on α-dicarbonyl compounds: (**A**) salt; (**B**) nitrite. Different letters above the bars indicate significant differences (n = 3, *p* < 0.05). Methylglyoxal (MGO), glyoxal (GO) and 3-deoxyglucuronide (3-DG).

**Figure 6 foods-12-00394-f006:**
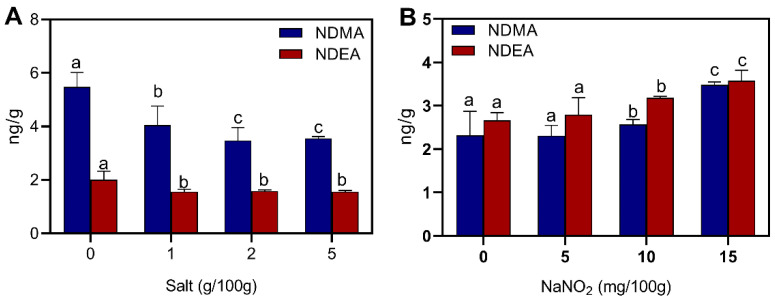
Effect of additives on NDMA and NDEA: (**A**) salt; (**B**) nitrite. Different letters above the bars indicate significant differences (n = 3, *p* < 0.05). N-nitrosodimethylamine (NDMA) and N-nitrosodiethylamine (NDEA).

**Figure 7 foods-12-00394-f007:**
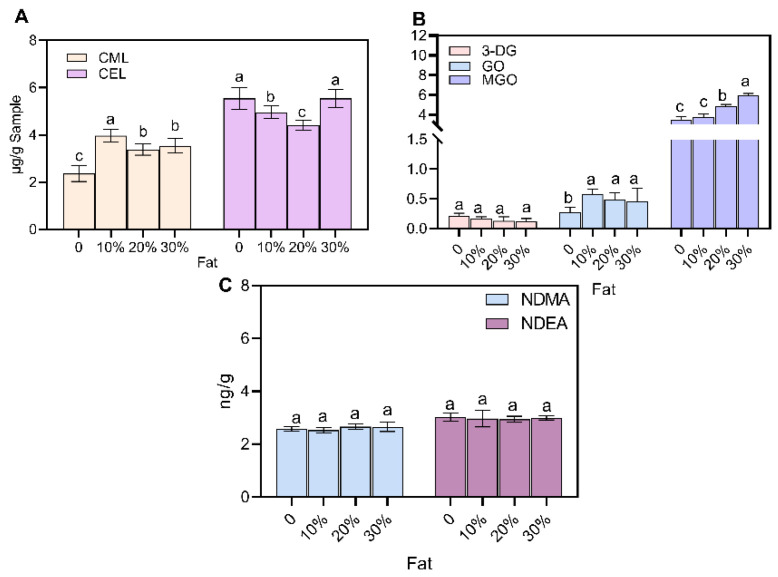
Effect of different ratios of fat addition on AGEs, α-dicarbonyl compounds and NAs: (**A**) AGEs; (**B**) α-dicarbonyl compounds; (**C**) NAs. Different letters above the bars indicate significant differences (n = 3, *p* < 0.05). Advanced glycation end products (AGEs), N-nitrosamines (NAs), N^ɛ^-carboxymethyl lysine (CML), N^ɛ^-carboxyethyllysine (CEL), methylglyoxal (MGO), glyoxal (GO), 3-deoxyglucuronide (3-DG), N-nitrosodimethylamine (NDMA) and N-nitrosodiethylamine (NDEA).

**Figure 8 foods-12-00394-f008:**
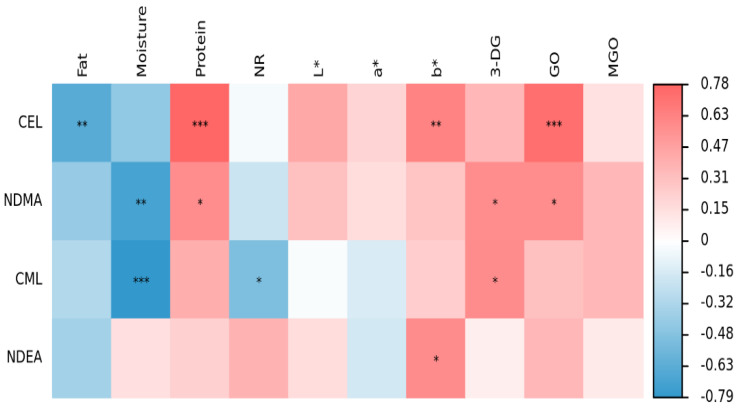
Correlation analysis: * indicates significant; ** indicates highly significant; *** indicates most significant. N^ɛ^-carboxymethyl lysine (CML), Nɛ-carboxyethyllysine (CEL), methylglyoxal (MGO), glyoxal (GO), 3-deoxyglucuronide (3-DG), N-nitrosodimethylamine (NDMA), N-nitrosodiethylamine (NDEA) and nitrite residue (NR).

**Table 1 foods-12-00394-t001:** Experimental conditions.

Temperature (°C)	Time (h)	Salt (%)	Nitrite (mg/100 g)	Fat (%)
70, 90, 110 and 130	2	2	10	5
110	1, 2, 3 and 4	2	10	5
110	2	0, 0.5, 1, 2 and 5	10	5
110	2	2	0, 5, 10 and 15	5
110	2	2	10	0, 10, 20 and 30

**Table 2 foods-12-00394-t002:** Proximate composition content during sausage processing.

	Fat (%)	Moisture (%)	Protein (%)	Nitrite Residue (mg/100 g)
Temperature (°C)				
70	3.07 ± 0.06 ^a^	64.20 ± 0.21 ^a^	38.57 ± 0.34 ^c^	6.57 ± 0.09 ^b^
90	1.90 ± 0.02 ^b^	61.60 ± 0.56 ^b^	37.17 ± 0.24 ^c^	9.09 ± 0.12 ^a^
110	1.47 ± 0.06 ^b^	60.26 ± 0.07 ^b^	44.12 ± 0.22 ^b^	8.12 ± 0.12 ^a^
130	0.55 ± 0.05 ^c^	55.48 ± 0.69 ^c^	51.47 ± 0.10 ^a^	4.67 ± 0.06 ^c^
Time (h)				
1	2.77 ±0.15 ^a^	66.42 ± 0.07 ^a^	44.38 ± 0.46 ^a^	5.93 ± 0.06 ^b^
2	1.51 ± 0.02 ^b^	61.35 ± 0.35 ^b^	37.42 ± 0.38 ^b^	9.10 ± 0.05 ^a^
3	0.78 ± 0.04 ^c^	59.71 ± 0.40 ^b^	43.29 ± 0.88 ^a^	4.82 ± 0.12 ^b^
4	0.70 ± 0.03 ^c^	52.72 ± 0.37 ^c^	43.50 ± 0.43 ^a^	5.08 ± 0.05 ^b^
Salt (g/100 g)				
0	3.27 ± 0.13 ^c^	52.95 ± 1.19 ^c^	39.94 ± 0.82 ^a^	4.42 ± 0.21 ^a^
0.5	3.43 ± 0.12 ^c^	55.11 ± 1.22 ^c^	40.48 ± 0.65 ^a^	4.56 ± 0.13 ^a^
1	4.39 ± 1.46 ^b^	57.36 ± 1.83 ^b^	40.68 ± 1.02 ^a^	4.67 ± 0.06 ^a^
2	5.51 ± 1.05 ^a^	59.99 ± 1.54 ^a^	41.12 ± 0.29 ^a^	4.72 ± 0.13 ^a^
5	3.99 ± 0.66 ^c^	58.16 ± 5.26 ^a^	40.56 ± 0.95 ^a^	4.65 ± 0.27 ^a^
Nitrite (mg/100 g)				
0	2.86 ± 0.05 ^a^	64.37 ± 0.14 ^a^	40.67 ± 0.56 ^a^	4.66 ± 0.09 ^a^
5	3.12 ± 0.04 ^b^	63.84 ± 0.05 ^a^	40.61 ± 0.26 ^a^	4.70 ± 0.12 ^a^
10	3.30 ± 0.02 ^b^	60.97 ± 6.09 ^b^	38.97 ± 0.93 ^b^	4.83 ± 0.05 ^a^
15	3.28 ± 0.10 ^b^	59.48 ± 0.07 ^b^	40.97 ± 0.38 ^a^	5.18 ± 0.16 ^a^

Results are mean ± standard error; the same lowercase letter in the same column indicates non-significant difference (*p* > 0.05), while different letters indicate a significant difference (*p* < 0.05).

**Table 3 foods-12-00394-t003:** Color values of sausages under different processing conditions.

Temperature (°C)	L*	a*	b*
70	55.49 ± 0.21 ^b^	8.36 ± 0.06 ^b^	10.13 ± 0.09 ^b^
90	62.40 ± 0.04 ^a^	11.57 ± 0.07 ^a^	10.58 ± 0.03 ^b^
110	64.00 ± 0.18 ^a^	11.13 ± 0.11 ^a^	12.20 ± 0.12 ^a^
130	64.07 ± 0.44 ^a^	11.65 ± 0.08 ^a^	13.23 ± 0.05 ^a^
Time (h)			
1	59.37 ± 0.05 ^b^	9.98 ± 0.11 ^b^	11.51 ± 0.47 ^a^
2	62.37 ± 0.15 ^a^	11.61 ± 0.12 ^a^	9.56 ± 0.23 ^b^
3	60.61 ± 0.05 ^b^	12.11 ± 0.79 ^a^	10.18 ± 0.45 ^b^
4	61.12 ± 0.10 ^b^	12.42 ± 0.66 ^a^	9.88 ± 0.16 ^b^
Salt (g/100 g)			
0	57.97 ± 1.16 ^c^	8.61±0.22 ^e^	10.99 ± 0.77 ^a^
0.5	59.48 ± 0.67 ^b^	9.10±0.08 ^d^	10.34 ± 0.20 ^a^
1	61.21 ± 0.58 ^a^	11.18±0.20 ^c^	10.18 ± 0.54 ^a^
2	62.23 ± 1.05 ^a^	12.34±0.06 ^b^	9.48 ± 0.27 ^a^
5	60.64 ± 3.91 ^b^	13.63±0.04 ^a^	9.97 ± 0.97 ^a^
Nitrite (mg/100 g)			
0	55.82 ± 0.46 ^d^	8.45 ± 0.21 ^c^	10.19 ± 0.11 ^b^
5	57.49 ± 0.17 ^c^	9.34 ± 0.05 ^c^	11.32 ± 0.96 ^a^
10	60.47 ± 1.65 ^b^	10.58 ± 0.88 ^b^	10.18 ± 0.47 ^b^
15	62.69 ± 1.84 ^a^	11.77 ± 0.70 ^a^	10.17 ± 0.28 ^b^

Results are mean ± standard error; the same lowercase letter in the same column indicates a non-significant difference (*p* > 0.05), while different letters indicate a significant difference (*p* < 0.05).

## Data Availability

The data used to support the findings of this study can be made available by the corresponding author upon request.

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
