# Peer review of "Advanced Glycation End Products and Nitrosamines in Sausages Influenced by Processing Parameters, Food Additives and Fat during Thermal Processing"

_foods, 2023, doi:10.3390/foods12020394_

Round 1

Reviewer 1 Report (Previous Reviewer 1)

Generally, the manuscript is improved considering the language, but the authors have focused on parts which were indicated before as particularly incorrect. It is necessary to correct the manuscript as a whole.

The authors have done some corrections and amendments to the manuscript in order to make it more clear and readable. However, still there are a lot of points that has to be improved. Mainly, my remarks are again related to the experimental plan and chapter 2.2 Preparation of sausage samples. Please, make the necessary corrections related to comments and observations given within the PDF version of the manuscript.

Author Response

Reviewer 2 Report (Previous Reviewer 4)

Dear Authors,

Great job! I have reviewed your remake of the manuscript and I'm satisfied with the revised form. All my comments were elaborated and necessary changes were made. 

Sincerely

Author Response

Thank you very much for your guidance and recognition

This manuscript is a resubmission of an earlier submission. The following is a list of the peer review reports and author responses from that submission.

Round 1

Reviewer 1 Report

This is a manuscript dealing with the analysis of AGEs, α-dicarbonyl compounds and NAs content in sausages as affected with thermal processing conditions (time and temperature) and formulation (content of salt, nitrites and fat). The results regarding the concentration of these harmful compounds in meat products (sausages) are very important. The study is good and the findings are satisfactory. However, the experimental design is not clear and it is very difficult to understand the relationship between the analyzed factors of the production process (time, temperature, fat, etc.).

The document is written in poor English, thus the language should be corrected by a native English speaker.

The chapter “2.2. Preparation of sausage samples” should be written again, providing description of clear and unquestionable experimental design (maybe an experimental scheme would be a good solution?).

Description of applied statistical analysis is missing. Also, provide a clear statement about the statistical significance of differences between presented values for all the Figures.

Discussion is simple and predictable. A large part of the text refers to the repetition of results that are already visible in tables and figures. Generally, this chapter should be corrected and amended.

Some specific comments and remarks are given within the PDF version of the manuscript.

Reviewer 2 Report

The manuscript is interesting as it brings the impact of processing parameters and AGE and nitrosamines production. 

The paper is well written, introduction has sufficient information about the topic, material and methods are detailed.

The results are well discussed and not just cited. The Tables and Figures are well formated and give support for the appropriate discussion.

In this context, I accept the manuscript in its current form.

Reviewer 3 Report

The Reviewer’s comments on paper Advanced glycation end products and nitrosamines in sausages influenced by processing parameters, food additives and fat during thermal processing; foods-1978706

There are several concerns that I have about this paper, and they are against its publication in the current form.

The first concern is about experimental design. In the M&M section, the authors stated that sausages were prepared with different levels of salt, nitrites and fat. Moreover, these sausages were heat treated at different temperatures during different times. How many factors were examined? Five? In 2.6 Statistical analysis section the procedure of the statistical analysis is missing. According to the R&D section, the effect of one factor was observed independently from the others. Why?

The second concern is about the number of replicates. According to the M&M section, only one replicate of the entire experiment was carried out. In that case, the numbers of samples analysed per treatment must be considered as pseudo replicates!

Reviewer 4 Report

Dear authors,

very nice paper, dealing with a huge problem in meat industry.

There are some suggestions:

line 48 - correct full name of NDEA

figure 1 - match upper and lower letters

line 99: were there repetitions of the product batches?

line 106 - correct fat ratios

line 335, 344, 345, 347 - correct "nitrite"